# Direct Probability Integral Method for Seismic Performance Assessment of Earth Dam Subjected to Stochastic Mainshock–Aftershock Sequences

**Weijie Huang [1], Yuanmin Yang [2], Rui Pang [2,3],* and Mingyuan Jing [2]**

[1] School of Civil and Architecture Engineering, Guangxi University, Nanning 530004, China; m15778112587@163.com
[2] State Key Laboratory of Coastal and Offshore Engineering, Dalian University of Technology, Dalian 116024, China; yangyuanmin@mail.dlut.edu.cn (Y.Y.); 22006113@mail.dlut.edu.cn (M.J.)
[3] School of Hydraulic Engineering, Faculty of Infrastructure Engineering, Dalian University of Technology, Dalian 116024, China
\* Correspondence: pangrui@dlut.edu.cn

**Abstract:** Studying the impact of mainshock–aftershock sequences on dam reliability is crucial for effective disaster prevention measures. With this purpose in mind, a new method for stochastic dynamic response analyses and reliability assessments of dams during seismic sequences has been proposed. Firstly, a simulation method of stochastic seismic sequences is described, considering the dependence between mainshock and aftershock based on Copula function. Then, a novel practical framework for stochastic dynamic analysis is established, combined with the improved point selection strategy and the direct probability integration method (DPIM). The DPIM is employed on a nonlinear system with one degree of freedom and compared with Monte Carlo simulation (MCS). The findings reveal that the method boasts exceptional precision and efficiency. Finally, the seismic performance of a practical dam was evaluated based on the above method, which not only accurately estimates the response probability distribution and dynamic reliability of the dam, but also greatly reduces the required calculations. Furthermore, the impact of aftershocks on dam seismic performance is initially evaluated through a probability approach in this research. It is found that seismic sequences will significantly increase the probability of earth dam failure compared with sequences of only mainshocks. In addition, the influence of aftershocks on reliability will further increase when the limit state is more stringent. Specifically, the novel analysis method proposed in this paper provides more abundant and objective evaluation indices, providing a dynamic reliability assessment for dams that is more effective than traditional evaluation methods.

**Keywords:** stochastic mainshock–aftershock sequences; DPIM; Copula function; dynamic reliability; seismic performance assessment





## 1. Introduction

Earth dams have been widely used because of their low price, convenient materials and long history. Earthquakes have a destructive effect on earth dams and may cause direct damage to people's lives and property. Many scholars have studied the nonlinear seismic response of the dams from the perspectives of stress, displacement and failure [1–3].

However, these studies fail to account for the impact that aftershocks have on structures. Aftershocks may cause significant secondary damage to the structure, thus exacerbating the situation [4]. A series of aftershocks led to 196 damaged dams in Sichuan Province 16 days after the main 2008 Wenchuan earthquake, China [5]. The aftershocks caused landslides of varying degrees in some dams and land reclamation areas in the 2011 Tohoku earthquake, Japan [6]. The large-scale aftershocks caused landslides in more geotechnical structures in the 2015 Gorkha earthquake, Nepal [7]. In addition, large shaking table tests

have shown that aftershocks may further damage the dam [8]. Hence, it is critical to investigate how the seismic sequence affects the structure. Some interesting research [9–12] has revealed the possible serious consequences of aftershocks for buildings. These studies usually use incremental dynamic analysis or MCS to analyze the seismic uncertainty of reinforced concrete buildings and steel buildings, but there is little research on geotechnical structures. Shokrabadi et al. [13] evaluated the structural performance of ductile-reinforced concrete frames using as-recorded seismic sequences and pointed out that seismic demand and danger may be overestimated or underestimated by the seismic sequences. Wang et al. [14] conducted a study on the nonlinear dynamics of a dam–reservoir–foundation system under seismic activity and discovered that the direction of the earthquake affects the concrete damage propagation processes. Yu et al. [15] synthesized the recorded earthquakes into earthquake sequences for calculating the inelastic SDOF system and studied the influence of artificial seismic sequences on system collapse capacity. Pang et al. [16,17] employed multiple stripe analysis to assess the vulnerability of a high concrete face rockfill dam (CFRD) exposed to seismic sequences. The study concluded that the fragility of CFRDs would be amplified by aftershocks. In the field of performance-based seismic design, Dong et al. [18] proposed a framework for a probabilistic seismic performance assessment subjected to earthquake sequences and investigated the probabilistic direct loss of bridges under seismic hazard. Wen et al. [19] studied the effect of aftershocks on the fragility of structures under different limit states based on engineering demand parameters and intensity measures.

Nevertheless, the studies did not consider the interdependent relationship between mainshocks and aftershocks and only combined them from the time perspective [20]. In fact, aftershocks and mainshocks share the same focal mechanism [21]. Based on this mechanism and engineering sites, Wang and Shen [22,23] presented a physical function model of ground motions. This method ultimately revealed the origin of seismic randomness, which is different from the spectral representation method [24]. Additionally, with the improvement in structural safety design requirements, deterministic seismic analysis methods can not meet the demands. It is necessary to propose a reliability analysis method that is suitable for geotechnical structures. The Monte Carlo method has been found to have low efficiency for calculating structural reliability. To address this issue, Li et al. [25] developed a probability density evolution method (PDEM) that employs equivalent extreme events. However, a discretization scheme is necessary to solve the probability density evolution equation (PDEE) and obtain the finite difference solution. Different difference schemes may cause dissipation or dispersion, thereby decreasing the stability and accuracy of the PDEM. In response, Chen et al. [26,27] presented a unified and effective DPIM based on Dirac delta function smoothing. This method can directly solve the probability density integral equation (PDIE) without using a difference scheme [28,29].

It is worth mentioning that research on the stochastic dynamic response analysis of earth dams considering influence on aftershocks is rarely reported. To this end, this paper will propose a more efficient strategy to evaluate the reliability of the dams. In this paper, a point selection strategy is applied to lower the generalized F-discrepancy (GFD) to reduce the number of samples required by traditional methods. Some stochastic seismic sequences are produced using a generation model that utilizes the superposition technique of harmonic wave groups. The correlation between mainshocks and aftershocks is established through Copula functions. MCS are employed to verify the accuracy and efficiency of the improved DPIM in a single degree of freedom (SDOF) system. Furthermore, the generated stochastic seismic sequences and improved DPIM are applied to a practical dam to obtain the stochastic dynamic response results. This study first reveals the effect of aftershocks on dam reliability and then analyzes the variation law of this influence with the design demand. The flowchart of this paper is shown in Figure 1.

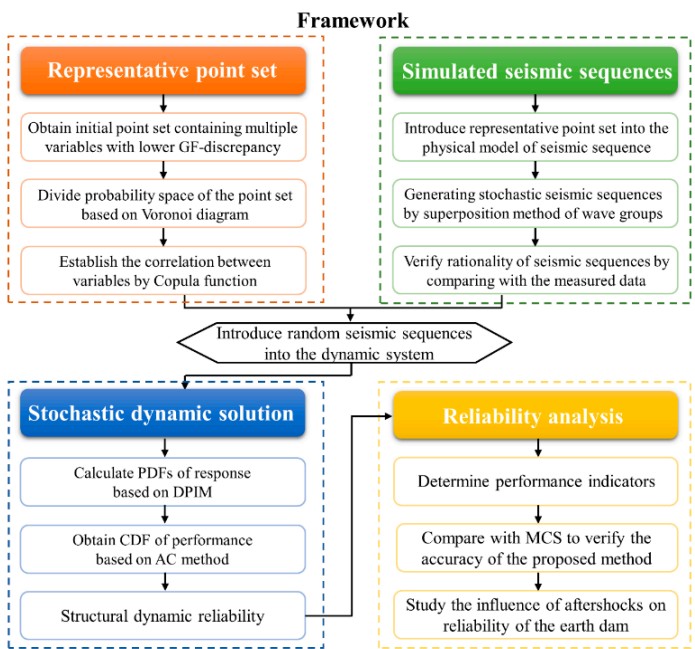

**Figure 1.** Flowchart of the framework of this study.

## 2. Simulation of Mainshock–Aftershock Sequences

The MCS method requires a large number of earthquake sequences to achieve precision, which poses difficulties in practice. A new method is required to accurately capture ground motion characteristics and generate multiple stochastic mainshock-aftershock sequences.

### 2.1. Physical Random Function Model of the Sequences

In seismic engineering, a physical random function model is often utilized to simulate ground motion [30]. The traditional method may not completely elucidate the underlying factors of randomness but instead focus on its manifestation. Wang et al. [23] considered that the randomness of ground motions is closely related to the generation mechanism of earthquakes and the seismic wave propagation mode, as shown in Figure 2.

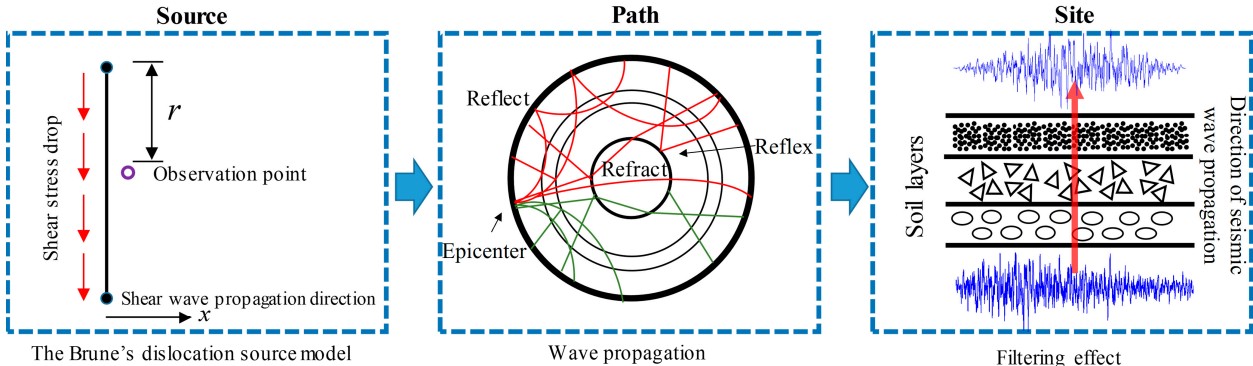

**Figure 2.** The establishment process of source–path–site mechanism.

Therefore, a model for ground motion, utilizing a physical random function based on the source–path–site mechanism, is proposed. This can be expressed as

$$a(t) = -\frac{1}{2\pi} \int_{-\infty}^{+\infty} A(\xi, \omega) \bullet \cos[\omega t + \Phi(\xi, \omega)] d\omega, \tag{1}$$

$$A(\xi,\omega) = \frac{A_0 \bullet \omega \bullet e^{-KR\omega}}{\sqrt{\omega^2 + (1/\tau)^2}} \bullet \sqrt{\frac{1 + 4\xi_g^2(\omega/\omega_g)^2}{\left[1 - (\omega/\omega_g)^2\right]^2 + 4\xi_g^2(\omega/\omega_g)^2}}, \tag{2}$$

$$\Phi(\xi,\omega) = \arctan(\frac{1}{\tau\omega}) - R \bullet \ln[a\omega + 1000b + 0.1323\sin(3.78\omega) + c\cos(d\omega)], \tag{3}$$

where $A(\xi,\omega)$ and $\Phi(\xi,\omega)$ are the Fourier amplitude spectrum and phase spectrum, respectively; $\omega$ is the circular frequency; $A_0$ is the source amplitude coefficient; $K$ is a coefficient to measure the attenuation effect of friction and $K = 10^{-5}$ s/km; $R$ is the epicentral distance; $\tau$ is Brune's coefficient; $\omega_g$ is the predominant circular frequency; $\xi_g$ is the equivalent damping ratio; $a, b, c, d$ are empirical parameters that comprehensively characterize the final waveform of synthetic ground motion.

The parameters in this model can be represented as random variables. From Equation (1), the physical model of the seismic sequence can be extended:

$$a(t) = \begin{bmatrix} a_M(t_1) \\ a_A(t_2) \end{bmatrix} = \begin{bmatrix} -\frac{1}{2\pi}\int_{-\infty}^{+\infty} A(\xi_M,\omega) \bullet \cos[\omega t_1 + \Phi(\xi_M,\omega)]d\omega \\ -\frac{1}{2\pi}\int_{-\infty}^{+\infty} A(\xi_A,\omega) \bullet \cos[\omega t_2 + \Phi(\xi_A,\omega)]d\omega \end{bmatrix}, \tag{4}$$

where, in the model of the mainshock and aftershocks, the time vectors for each event are denoted as $t_1, t_2$, while the random variables are represented by $\xi_M, \xi_A$ ($M$ and $A$ represent the mainshocks and aftershocks).

Therefore, the model parameters of the seismic sequences (Equation (4)) can be summarized as:

$$\xi = [\xi_M, \xi_A] = \left[A_{0M}, \tau_M, a_M, b_M, c_M, d_M, A_{0A}, \tau_A, a_A, b_A, c_A, d_A, \xi_g, \omega_g\right], \tag{5}$$

where the variables $A_0$ and $\tau$ represent the source randomness. $a, b, c, d$ represent randomness in the propagation process. Randomness at the local site is described by variables $\xi_g$, $\omega_g$. Tables 1 and 2 present the probability distribution and statistical features of $\xi$ [22,31,32].

**Table 1.** Distribution and statistics of seismic characteristics.

| Variable | Mainshocks | | | Aftershocks | | |
|---|---|---|---|---|---|---|
| | Type | $\mu$ | $\sigma$ | Type | $\mu$ | $\sigma$ |
| $A_0$ | Lognormal | −2.85 | 1.26 | Lognormal | −4.54 | 1.32 |
| $\tau$ | Lognormal | −1.75 | 1.24 | Lognormal | −2.15 | 1.71 |
| $a$ | Lognormal | 1.32 | 0.56 | Lognormal | 1.83 | 0.49 |
| $b$ | Lognormal | 1.89 | 0.55 | Lognormal | 1.95 | 0.50 |
| $c$ | Weibull | 1.33 | 1.22 | Weibull | 1.76 | 1.66 |
| $d$ | Weibull | 1.23 | 1.20 | Weibull | 1.44 | 1.35 |

**Table 2.** Distribution and statistics of site characteristics.

| Variable | Type | $\mu$ | $\sigma$ |
|---|---|---|---|
| $\xi_g$ | Weibull | 0.49 | 4.674 |
| $\omega_g$ | Weibull | 12.63 | 1.26 |

### 2.2. Establishment of the Mainshocks and Aftershocks Correlation

In this paper, the factors affecting the characteristics of ground motions were abstracted into variables. Thus, the relationship between the mainshocks and aftershocks can be established by the correlation between variables $\xi_M$ and $\xi_A$, and the Copula functions between variables are listed in Table 3 [22,31,32].

**Table 3.** The best Copula type and parameter.

| Best Copula | $C_{A_{0M}A_{0A}}$ | $C_{\tau_M\tau_A}$ | | | $C_{a_Ma_A}$ | $C_{b_Mb_A}$ | $C_{c_Mc_A}$ | $C_{d_Md_A}$ | |
|---|---|---|---|---|---|---|---|---|---|
| | Plackett | Student | | | Plackett | Clayton | Clayton | Student | |
| Parameters | $\theta$ | $\theta$ | $\nu$ | | $\theta$ | $\theta$ | $\theta$ | $\theta$ | $\nu$ |
| | 20.172 | 0.233 | | 4 | 3.199 | 1.330 | 0.523 | 0.212 | 4 |

Figure 3 depicts the final set of dependent representative points on the projected subspace. It is evident that this method produces dependent variables with a high level of realism and strong nonlinearity. This correlation cannot be constructed only by virtue of the correlation coefficient.

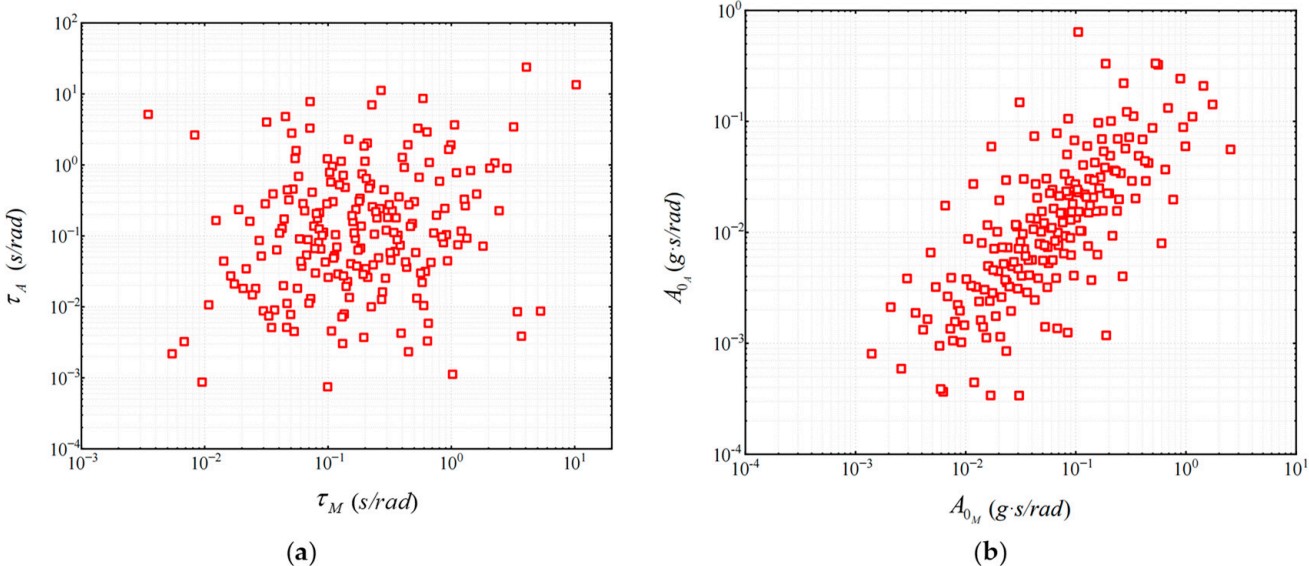

(a)                         (b)

**Figure 3.** Dependent representative point set. (**a**) Relationship between variables $A_{0_M}$ and $A_{0_A}$ (**b**) Relationship between variables $\tau_M$ and $\tau_A$.

The principle of generating dependent random variables based on the Copula function is complex, and this method will be introduced in detail in Section 3.

### 2.3. Ground Motions Generation

The inverse Fourier transform is often utilized to generate ground motions. However, it is important to note that the fine structure of the ground motion phase spectrum can have a significant impact on the resulting waveform. As such, the ground motion generated based on this method cannot accurately reflect the nonstationary nature of ground motion. Due to the wave group properties of seismic waves, it is more reasonable to use the superposition method of narrow-band harmonic wave groups. The specific technical flow of this method is shown in Figure 4. Firstly, the amplitude and phase spectra of seismic waves are transformed into the changes in the amplitude and phase of each wave group through the inverse Fourier transform method. The wave group can then be superimposed on a specific local site, and a simulated ground motion time history with nonstationary characteristics can be produced.

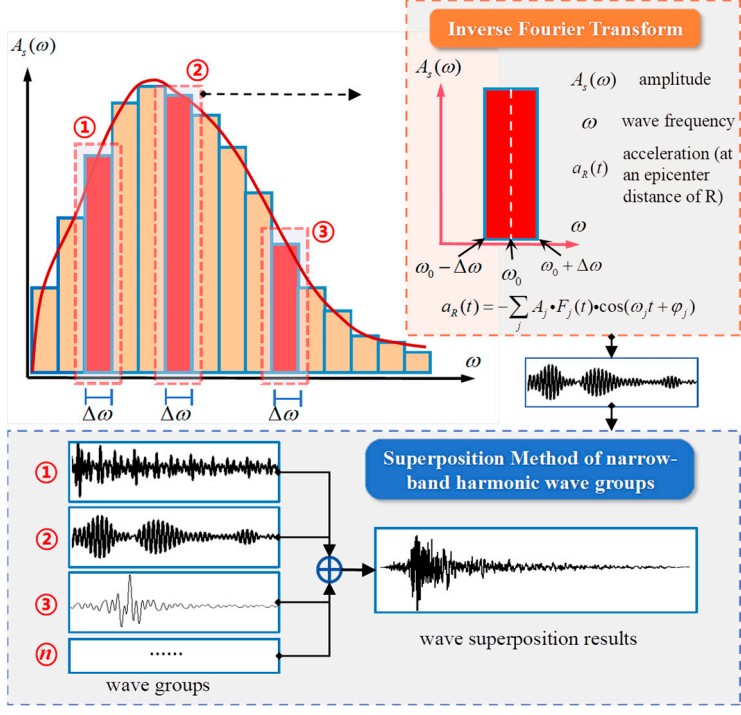

**Figure 4.** Schematic diagram of ground motion simulation.

The specific mathematical calculations corresponding to the procedure are as follows.

The amplitude spectrum is divided into uniform narrow bands, and each narrow band is assumed to be constant. A time history of acceleration on the site with epicentral distance $R$ can be expressed as:

$$a_R(t) = -\sum_j A_j \bullet F_j(t) \bullet \cos(\omega_j t + \varphi_j), \tag{6}$$

$$A_j = \frac{2}{\pi} \frac{A_0 \omega_j e^{-KR\omega_j}}{\sqrt{\omega_j^2 + (1/\tau)^2}} \bullet \sqrt{\frac{1 + 4\xi_g^2(\omega_j/\omega_g)^2}{\left[1 - (\omega_j/\omega_g)^2\right]^2 + 4\xi_g^2(\omega_j/\omega_g)^2}}, \tag{7}$$

$$F_j(t) = \frac{\sin\left[(t - \frac{x}{c_j})\Delta\omega_j\right]}{(t - \frac{x}{c_j})}, \tag{8}$$

$$c_j = \frac{\mathrm{d}\omega}{\mathrm{d}k}\Big|_{\omega=\omega_j} = \frac{a\omega_j + 1000b + 0.1323\sin(3.78\omega_j) + c\cos(d\omega_j)}{d(a + \cos^2(d\omega_j))}, \tag{9}$$

$$\varphi_j = \arctan(\frac{1}{\tau\omega_j}) - R\bullet\ln[a\omega_j + 1000b + 0.1323\sin(3.78\omega_j) + c\cos(d\omega_j)], \tag{10}$$

where the $j$-th wave group's amplitude is denoted as $A_j$, with $F_j(t)$ and $\varphi_j$ being the time energy envelope function and the phase. The group velocity of the j-th wave group is denoted by $c_j$.

In summary, seismic sequences of any number can be generated based on the stochastic seismic sequence physical model of Equation (4) and the superposition method of Equation (6).

### 2.4. Verification of Stochastic Seismic Sequences

This chapter presents the generation of 140 seismic sequence groups (PGA = 0.25 g), with four examples displayed in Figure 5. The ground motion exhibits nonstationary

characteristics, with instances where the aftershock's amplitude surpasses that of the mainshock, mirroring realistic seismic events.

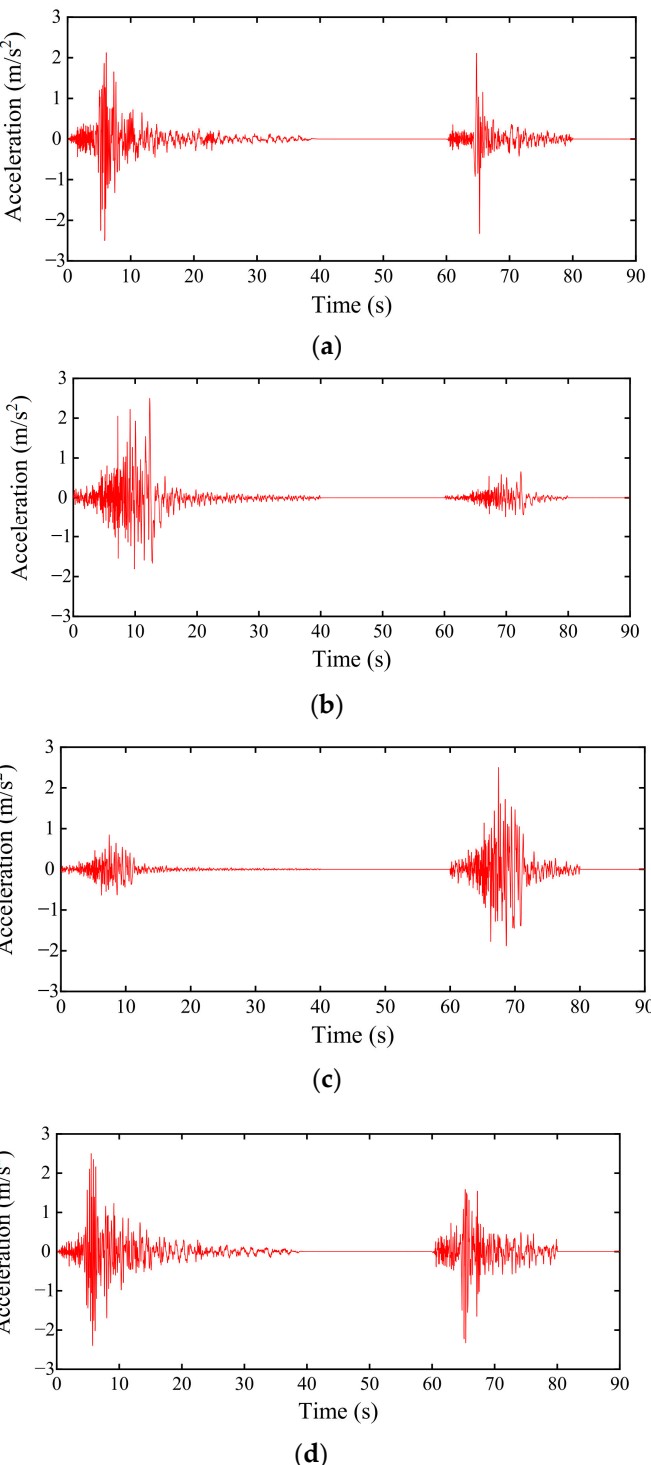

**Figure 5.** Four examples of acceleration time history of stochastic earthquake sequences. (**a**) Sample #1 of acceleration time history of seismic sequence. (**b**) Sample #2 of acceleration time history of seismic sequence. (**c**) Sample #3 of acceleration time history of seismic sequence. (**d**) Sample #4 of acceleration time history of seismic sequence.

A total of 636 groups of real data were used for comparison with the simulation results to verify the accuracy of the seismic generation method proposed in this paper. Figure 6a,b

present a comparison of the mean and standard variance of stochastic ground motion with the measured earthquake sequences. The results indicate that the ground motions generated in this study have favorable statistical characteristics. Additionally, Figure 6c depicts the comparison between the mean value of the response spectrum of 140 stochastic ground motions and the measured seismic response spectrum. The comparison suggests that the method proposed in this paper is reliable and accurate in generating realistic seismic ground motions.

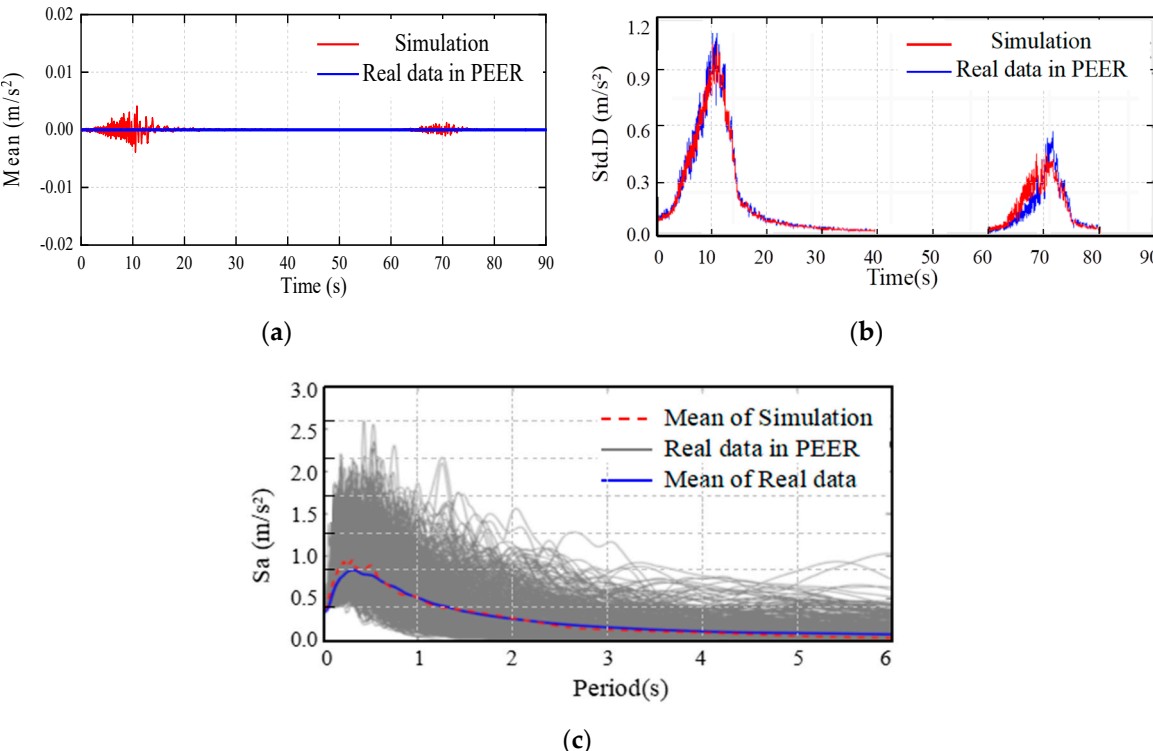

**Figure 6.** Comparison between simulation data and real data of stochastic seismic sequences. (**a**) Comparison of mean values. (**b**) Comparison of standard deviation. (**c**) Comparison of acceleration response spectrum.

## 3. Direct Probability Integral Method

The MCS method cannot reflect the various degrees of freedom of the earth dam and long duration of seismic events. Therefore, an improved DPIM is introduced to calculate the stochastic dynamic response of dams. Additionally, to achieve accurate results, this paper proposes an enhanced selection strategy for representative point sets, which significantly decreases the number of representative points.

### 3.1. Probability Density Integral Equation

Chen and Yang [27] devised the DPIM to address the probability information of the structural stochastic response and derive the dynamic reliability. Based on the principle of the conservation of probability, this approach holds that random factors are transferred from random input vectors to random output vectors in a given system, and the propagation process can be expressed as a mapping:

$$\mathbf{Y}(t) = g[\mathbf{X}(\mathbf{\Theta}, t)] = g(\mathbf{\Theta}, t), \tag{11}$$

where $g(\cdot)$ is a unified mapping function; $\mathbf{X}(\mathbf{\Theta}, t)$ is a stochastic process that denotes excitation here; and $\mathbf{\Theta}$ is a random factor describing the randomness of the system.

The PDIE of output function **Y** can be expressed as

$$p_{\mathbf{Y}}(\mathbf{y}, t) = \int_{\Omega_{\Theta}} p_{\Theta}(\theta) \delta[\mathbf{y} - g(\theta, t)] \mathrm{d}\theta, \tag{12}$$

where $\delta(\cdot)$ is the Dirac delta function; $p_{\Theta}(\theta)$ is the probability density function of $\Theta$; and **y** is the output vector, which represents the response. Since the integral Equation (12) is difficult to solve directly, the smoothing technique of replacing the discontinuous Dirac function with the continuous Gaussian function was an effective simplification method for the equation. Equation (13) is further obtained by reducing the standard deviation of the Gaussian function to nearly zero.

$$\lim_{\sigma \to 0} p_N(\mathbf{y}; \mu, \sigma) = \lim_{\sigma \to 0} \frac{1}{\sqrt{2\pi}\sigma} e^{-(\mathbf{y}-\mu)^2/2\sigma^2} = \delta(\mathbf{y} - \mu), \tag{13}$$

where $\sigma$ is a smoothing parameter and $\sigma = \Delta y$ ($\Delta y$ is the discretized step length of **y**) was adopted in a simplified way.

Therefore, the PDIE can be conveniently solved by incorporating Equation (13) into Equation (12):

$$\hat{p}_{\mathbf{Y}}(\mathbf{y}, t) = \sum_{q=1}^{N} \left\{ \frac{1}{\sqrt{2\pi}\sigma} e^{-[\mathbf{y}-g(\theta_q, t)]^2/2\sigma^2} P_q \right\}, \tag{14}$$

where $P_q$ is the probability carried by random representative points, which can be expressed as

$$P_q = \int_{\Omega_q} p_{\Theta}(\mathbf{x}) \mathrm{d}\mathbf{x}. \tag{15}$$

### 3.2. Dynamic Reliability Analysis Based on Absorbing Condition Method

To simplify the dynamic problem, it is necessary to split the probability information of a dynamic system. Hence, the probability of a representative point $P_q$, represented by the $q$-th response can be decomposed into two parts: the failure component $P_{q,f}(t)$ and the survival component $P_{q,s}(t)$, namely,

$$P_q(t) = P_{q,f}(t) + P_{q,s}(t). \tag{16}$$

The performance function is generally used to describe the transformation of the structure from normal state to failure state under load, which can be mathematically expressed as:

$$Z(t) = \overline{Y} - Y(\Theta, t), \tag{17}$$

where $\overline{Y}$ is the prescribed limit state or threshold.

Therefore, the performance function $Z \leq 0$ serves as a reflection of the failure event. The $q$-th assigned probability of the failure part is expressed as

$$P_{q,f}(t_i) = 0, \quad z \in \Omega_{Z,f} = \left\{ z \big| g(\theta_q, t_i) \leq 0 \right\}. \tag{18}$$

In order to maintain the conservation of probability in the system, the probability distribution that remains must be substituted by the survival portion. Therefore, the probability density function (PDF) used to solve the stochastic response in the safe domain, combined with PDIE (Equation (14)), can be expressed as

$$\hat{p}_{Z,s}(z, t) = \sum_{q=1}^{N} \left\{ \frac{1}{\sqrt{2\pi}\sigma} e^{-[z-g(\theta_q, t)]^2/2\sigma^2} P_q(t) \right\}. \tag{19}$$

The first passage reliability of the dynamic system can be quantified as

$$P_s(t) = \Pr[Z > 0] = \int_0^\infty p_{Z,s}(z,t)\mathrm{d}z = \sum_{j=1}^{N_z} \hat{p}_{Z,s}(z,t)\Delta z, \tag{20}$$

where $N_z = z/\Delta z$.

### 3.3. Selection Point for Dependent Random Variables

In order to effectively apply DPIM to stochastic dynamical systems, it is imperative to introduce a random factor $\Theta$ and assign a probability $P_q$ to it. This introduction process is actually the process of random representative point set selection and partition of probability space. In this paper, a point in the representative point set represents a random variable $\xi$, which can generate a seismic sequence. The 140 groups of seismic sequences in Section 2 need a representative point set containing 140 points, and each point has 14 dimensions.

The accuracy and efficiency of the DPIM are affected by the point set's cardinality and uniformity. Thus, we propose the concept of GFD to measure point set uniformity. The complex nonlinear multidimensional random variable set involving non-uniform and non-Gaussian distributions can be represented by reducing GFD [33]. The GFD can be expressed as

$$D_{GF} = \max_{1 \le i \le s}\{\sup|F_{n,i}(x) - F_i(x)|\}, \tag{21}$$

where $F_{n,i}(x)$ is the empirical marginal CDF, which is computed using Equation (22); $F_i(x)$ is the marginal CDF of the $i$-th random variable.

$$F_{n,i}(x) = \sum_{q=1}^n P_q \bullet I\{x_{q,i} \le x\}, \tag{22}$$

where $x_{q,i}$ is the $i$-th element of random variable $\mathbf{x}_q$; $I\{\cdot\}$ is the indicator function, $I\{A\} = 1$, if and only if event $A$ holds true; otherwise, $I(A) = 0$.

The s-dimensional initial independent scattered point set $x_q = (x_{q,1}, x_{q,2}, \cdots, x_{q,s})$, $q = 1, 2, \ldots, n$ can be acquired through various sampling methods, such as the Sobol set. However, a point set with a lower GFD can be attained by adjustment with the following two equations:

$$x_{m,i}^* = F_i^{-1}(\sum_{q=1}^n \frac{1}{n} \bullet I\{x_{q,i} < x_{m,i}\} + \frac{1}{2} \bullet \frac{1}{n}), \tag{23}$$

$$x_{m,i}^{**} = F_i^{-1}(\sum_{q=1}^n P_q \bullet I\{x^{**}{}_{q,i} < x^{**}{}_{m,i}\} + \frac{1}{2} \bullet P_m), \tag{24}$$

Copula theory [34] allows for the conversion of several marginal distributions into a multivariate joint distribution using a Copula function. Recently, Copula theory has been increasingly applied in the fields of bridge engineering [35,36], geotechnical engineering [37,38], and hydrology [39]. Sklar's theorem states that every multivariate distribution F with marginal distribution $F_1(x_1), \ldots, F_n(x_n)$ can be expressed as

$$\overline{F}(x_1, \ldots, x_n) = C\{F_1(x_1), \ldots, F_n(x_n)\}. \tag{25}$$

Therefore, the marginal conditional distributions can be expressed as

$$\overline{F}(x_1|x_2) = \frac{\partial C_{x_1 x_2}\{F_1(x_1), F_2(x_2)\}}{\partial F_2(x_2)}. \tag{26}$$

Now, one can establish the correlation between $x_1^{**}$ and $x_2^{**}$ in point set $x^{**}$ by copula function $C\{\cdot\}$:

$$x_2^c = \overline{F}^{-1}(x_2^{**}|x_1^{**}), \tag{27}$$

where $\overline{F}^{-1}(\cdot)$ is the inverse function of $\overline{F}(x_1|x_2)$ in Equation (26); $x_2^c$ is a random variable with correlation with $x_1^{**}$, which is transformed based on variable $x_2^{**}$. The six other pairs of dependent variables in $x^{**}$ adopt the same transformation.

In Section 2, random factors were abstracted into random variables. Generating a large number of sample points conform to a specific distribution is an efficient way to conduct a random dynamic analysis of structures. Thus, through the method in this section, the required sample points can be obtained. Figure 7 displays the scatter plot of a Sobol set of 140 points in the projected subspace of high-dimensional variables.

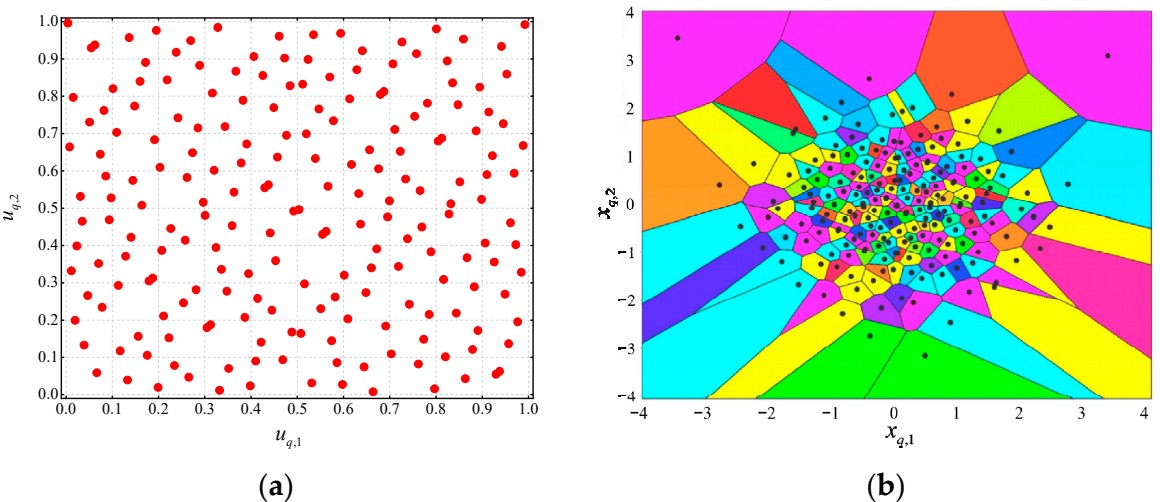

**Figure 7.** Sampling points on the projected subspace. (**a**) Initial independent scattered points. (**b**) Independent representative point set (Color is only for distinguishing).

The process from Figure 7a to Figure 7b represents the process by which independent sampling points are assigned probabilities in a high-dimensional space. Where Figure 7a depicts the initial independent scattered points, Figure 7b shows the Voronoi cells of the independent representative point set, with polygons colored to reflect the assigned probability $P_q$ of this point. A larger polygon area indicates a higher probability of distribution. The horizontal and vertical axes of the graph are used to represent the coordinates of the position of the point in space.

### 3.4. Novel Probabilistic Analysis and Evaluation Framework

The numerical procedure for a reliability analysis of structures subjected to seismic sequences is shown in Figure 8a. The reason why the proposed method is more efficient is that it forms a perfect mapping relationship between the two parts to achieve dimension reduction, as shown in Figure 8b.

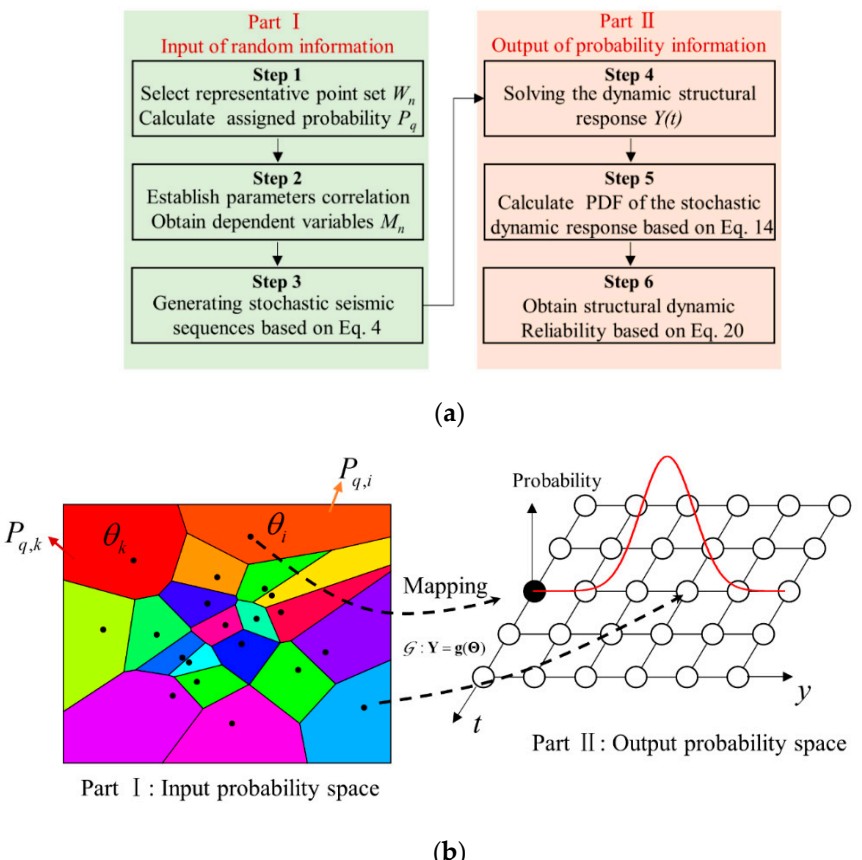

(a)

(b)

**Figure 8.** Principle of stochastic dynamic analysis. (**a**) Flowchart of dynamic reliability analyses. (**b**) Evolution of probability information (Color is only for distinguishing).

### 3.5. Verification of DPIM

The MCS method is applied to an SDOF system to verify the accuracy of the probability method. To validate the proposed stochastic dynamic analysis method in this paper, a seismic sequence–excitation nonlinear SDOF system (depicted in Figure 9) was introduced.

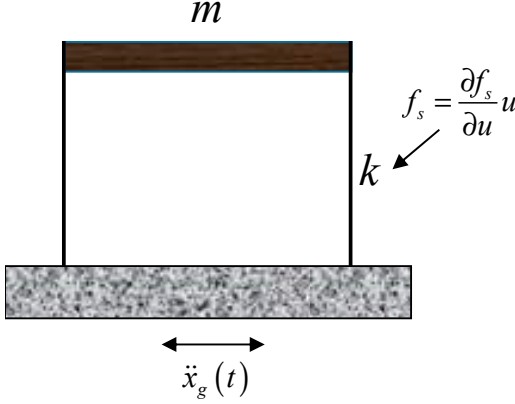

**Figure 9.** Nonlinear SDOF dynamic system.

The dynamic balance equation of an SDOF system subjected to ground motion can be expressed as follows:

$$m\ddot{u} + c\dot{u} + f(u) = -m\ddot{x}_g(t), \tag{28}$$

where $m$ and $c$ are the mass and damping, respectively; $f$ is the nonlinear restoring force in the form of $f = \frac{\partial f}{\partial u} u$; $\ddot{x}_g(t)$ is the acceleration of ground motion.

This paper adopts the bilinear elastoplastic model, in which the yield force is 160 kN and the restoring force hysteresis loop is shown in Figure 10. $m = 1$ kN, $c = 6$ kN·s/cm and initial stiffness $k = 10^6$ kN/m are used. The Newmark-β method ($\beta = 0.25$) is used to solve the physical equation in Equation (28).

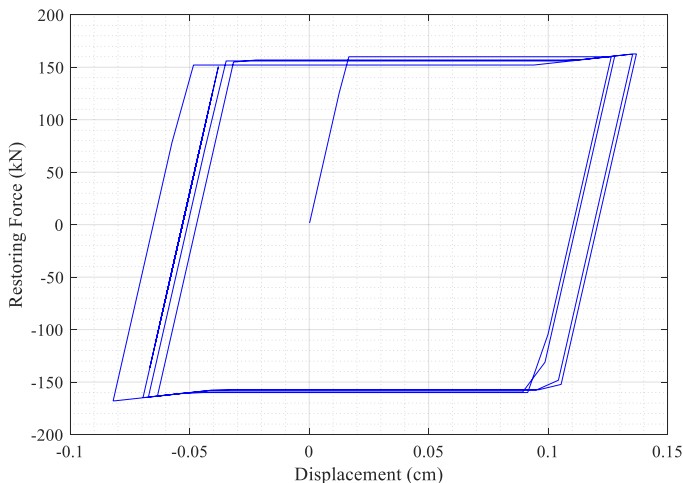

**Figure 10.** Restoring force hysteresis loop.

The probability information and reliability of the SDOF system can be obtained through the process shown in Figure 8. Furthermore, the effectiveness of the proposed method was verified on an SDOF structure by utilizing 50,000 Monte Carlo samples. The obtained results were compared and found to closely align with those obtained through the proposed method.

Figure 11 illustrates the PDF of the absolute displacement of the SDOF at representative times, indicating the accuracy of the proposed method. Although there is a slight gap in the details, compared with the large amount of calculation required by MCS, these errors can be completely ignored.

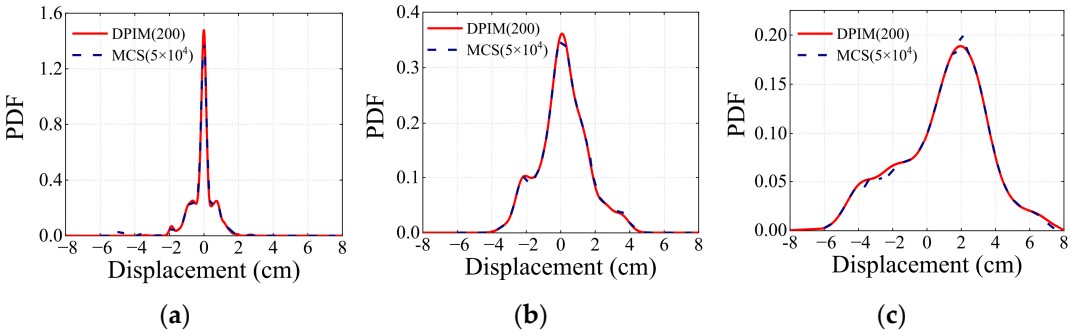

**Figure 11.** The PDF of the displacement of SDOF at representative times. (**a**) PDF at 10 s. (**b**) PDF at 50 s. (**c**) PDF at 90 s.

Figure 12 displays the CDF of the displacement's absorbing condition (AC), while Figure 13 shows the dynamic reliability of an SDOF system under various thresholds. The calculation results, errors, and computation time for this analysis are detailed in Table 4. The proposed probability analysis method demonstrates a comparable reliability to MCS and offers superior calculation efficiency.

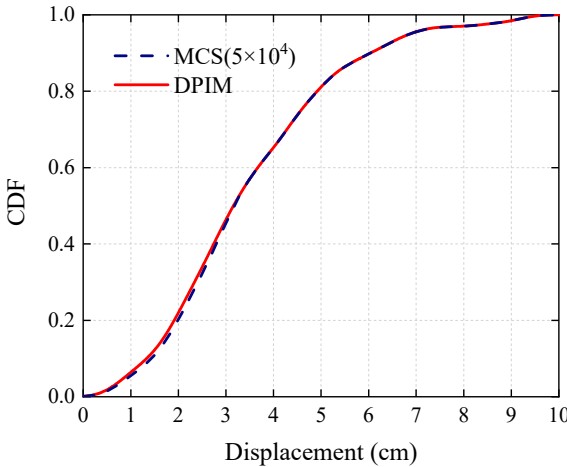

**Figure 12.** The CDF of the absorbing condition of displacement.

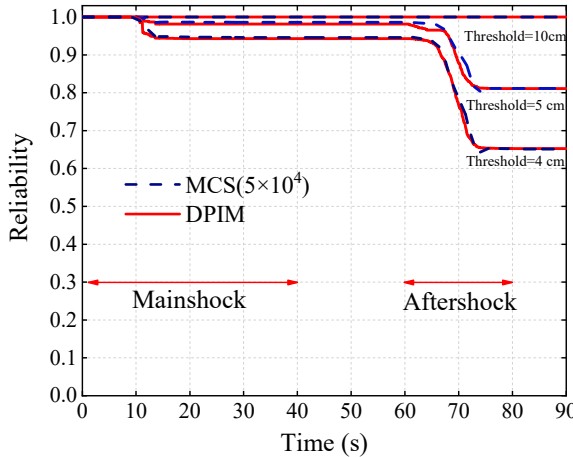

**Figure 13.** Dynamic reliability of SDOF under various thresholds.

**Table 4.** Errors and calculation time.

| Methods | Sample Size | Errors | Calculation Time (s) |
|---|---|---|---|
| DPIM | 200 | 0.10% | 4.51 |
| MCS | $5 \times 10^4$ | - | 1059.04 |

## 4. Dynamic Reliability Analysis of an Earth Dam Subjected to Stochastic Seismic Sequences

This section aims to present a stochastic dynamic analysis and reliability evaluation of the considered dam based on DPIM. We study the effects of aftershocks on the dam through both deterministic and stochastic dynamic analysis. Moreover, we discuss, for the first time, the impact of aftershocks on dam reliability based on our proposed analysis framework.

### 4.1. Finite Element Model and Material Parameters

The present study is based on an asphaltic–concrete–core rockfill dam named Quxue, which is located in Sichuan Province in China. The dam is 220 m long and 164 m high, and the dam crest is 15 m wide. The full reservoir level is equal to 160 m and the main cross-section is presented in Figure 14. The seismic fortification intensity is VIII, and the epicentral distance is 18.5 km. The finite element model is given in Figure 15.

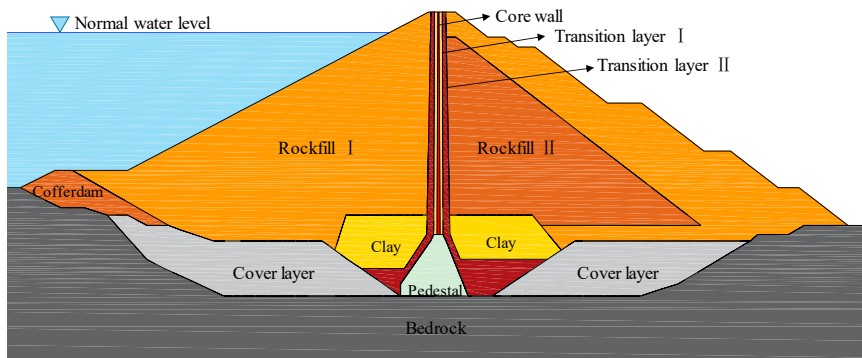

**Figure 14.** Main cross-section of the studied dam (1:1000).

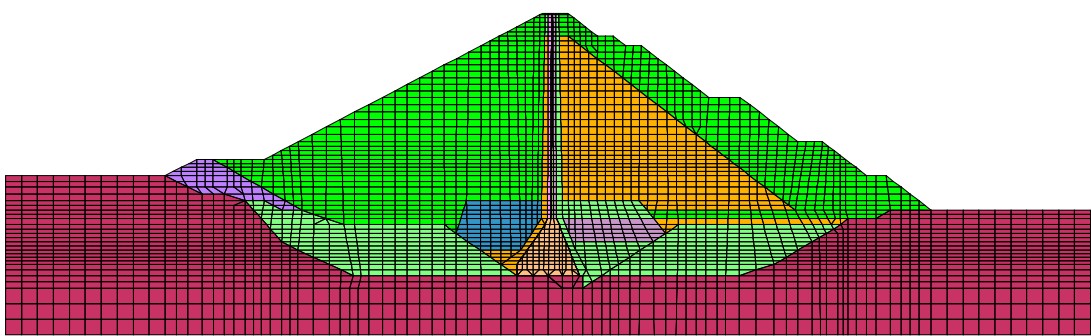

**Figure 15.** Finite element model of the dam (1:1000) (Different colours represent different material partitions).

The viscoelastic boundary element is adopted to simulate the infinite foundation, and the seismic wave is equivalent to the load applied to the boundary to simulate the traveling wave effect of the seismic wave [40]. The unrecoverable plastic deformation of structures is the main factor leading to the failure of geotechnical structures. Therefore, the generalized plastic P–Z constitutive model [41–43] is used to effectively simulate the dilatancy, shear shrinkage and cyclic cumulative deformation of soil under dynamic loads. This can be expressed as

$$d\sigma = \mathbf{D}^{ep} : d\varepsilon, \tag{29}$$

where $\mathbf{D}^{ep}$ is an elastic–plastic matrix, which can be expressed as

$$\mathbf{D}^{ep} = \mathbf{D}^{e} - \frac{\mathbf{D}^{e} : \mathbf{n}_{gL/U} \otimes \mathbf{n} : \mathbf{D}^{e}}{H_{L/U} + \mathbf{n} : \mathbf{D}^{e} : \mathbf{n}_{gL/U}}, \tag{30}$$

where $\mathbf{n}_{gL}$, $\mathbf{n}_{gU}$ are the directions of the loading plastic flow and unloading plastic flow, respectively; $\mathbf{n}$ is the vector of the loading direction; and $H_L$, $H_U$ are the plastic moduli of loading and unloading, respectively.

The increase in strain can be divided into elastic and plastic increments:

$$d\varepsilon_e = \mathbf{C}^{e} : d\sigma, \tag{31}$$

$$d\varepsilon_P = \frac{1}{H_{L/U}} \mathbf{n}_{gL/U} \otimes \mathbf{n} : d\sigma. \tag{32}$$

The main material parameters are determined by the dynamic triaxial test. These are given in Tables 5–8, and their accuracy was calibrated in the report [44]. The GEODYNA software served as the tool for both static and dynamic calculations. Due to the existence of an upstream water level and core wall, zone I and zone II of the dam are saturated and unsaturated, respectively, which was achieved by modifying the parameters [45–47].

**Table 5.** Rockfill parameters for generalized plastic model.

| $\rho$ (kg/m$^3$) | $G_0$ | $K_0$ | $M_g$ | $M_f$ | $\alpha_f$ | $\alpha_g$ | $H_0$ | $H_{U0}$ |
|---|---|---|---|---|---|---|---|---|
| 2250 | 1130 | 1440 | 1.75 | 1.60 | 0.35 | 0.35 | 1500 | 2000 |
| $m_s$ | $m_v$ | $m_l$ | $m_u$ | $r_d$ | $\gamma_{DM}$ | $\gamma_u$ | $\beta_0$ | $\beta_1$ |
| 0.40 | 0.40 | 0.20 | 0.30 | 180 | 70 | 7 | 35 | 0.038 |

**Table 6.** Transition parameters for generalized plastic model.

| $\rho$ (kg/m$^3$) | $G_0$ | $K_0$ | $M_g$ | $M_f$ | $\alpha_f$ | $\alpha_g$ | $H_0$ | $H_{U0}$ |
|---|---|---|---|---|---|---|---|---|
| 2365 | 1330 | 1540 | 1.70 | 1.40 | 0.20 | 0.32 | 1000 | 1800 |
| $m_s$ | $m_v$ | $m_l$ | $m_u$ | $r_d$ | $\gamma_{DM}$ | $\gamma_u$ | $\beta_0$ | $\beta_1$ |
| 0.20 | 0.20 | 0.25 | 0.10 | 120 | 50 | 7 | 30 | 0.021 |

**Table 7.** Core wall parameters for generalized plastic model.

| $\rho$ (kg/m$^3$) | $G_0$ | $K_0$ | $M_g$ | $M_f$ | $\alpha_f$ | $\alpha_g$ | $H_0$ | $H_{U0}$ |
|---|---|---|---|---|---|---|---|---|
| 2460 | 1000 | 800 | 1.70 | 1.60 | 0.45 | $-0.80$ | 800 | 2000 |
| $m_s$ | $m_v$ | $m_l$ | $m_u$ | $r_d$ | $\gamma_{DM}$ | $\gamma_u$ | $\beta_0$ | $\beta_1$ |
| 0.50 | 0.50 | 0.10 | 0.20 | 220 | 70 | 7 | 10 | 0.01 |

**Table 8.** Parameters for linear-elastic model.

| Material | $\rho$/(kg/m$^3$) | $E$/(GPa) | $\nu$ |
|---|---|---|---|
| Bed rock | 2650 | 13 | 0.250 |
| Concrete base | 2450 | 30 | 0.167 |

### 4.2. Dynamic Response of the Earth Dam

The dynamic responses of the dam after earthquakes can be obtained based on FEM. Among the 140 groups of calculation results, one group was selected to focus on the dynamic response of the dam after the seismic sequence. The horizontal deformation and settlement of the dam following the mainshock are shown in Figure 16. The maximum horizontal displacement is 27 cm, which occurs at the downstream slope of the dam. And the maximum settlement is 37.5 cm, occurring at the dam crest.

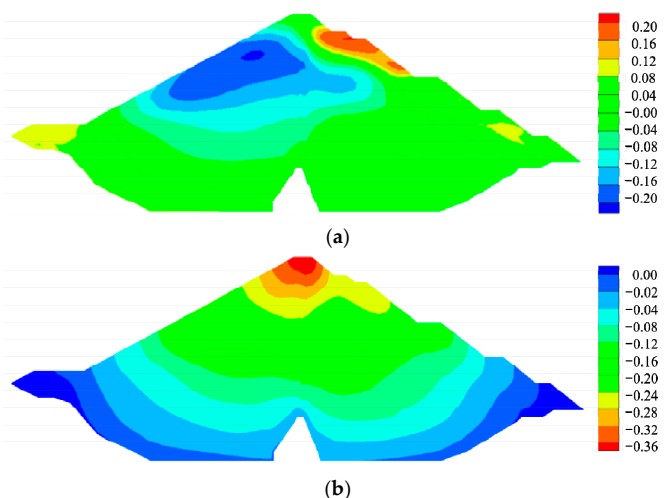

(a)

(b)

**Figure 16.** Dynamic response after mainshock of a single sample. (**a**) Horizontal deformation. (**b**) Vertical deformation.

The horizontal deformation and settlement of the dam following the aftershock are shown in Figure 17. The maximum horizontal displacement can reach 53 cm, which occurs at the downstream slope of the dam. And the maximum settlement can reach 72 cm, which occurs at the downstream dam crest.

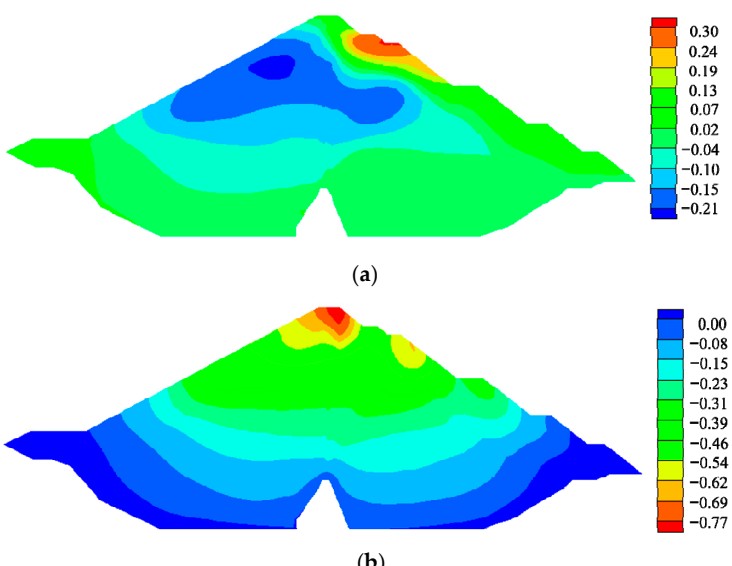

**Figure 17.** Dynamic response after aftershock of a single sample. (**a**) Horizontal deformation. (**b**) Vertical deformation.

Geotechnical materials with nonlinearity will undergo large residual deformations during earthquakes. Water will overflow the dam crest and cause serious disaster when the upstream water-retaining side of the dam is lower than the water level. Therefore, the displacement at the upstream dam crest needs special attention. Figure 18a displays the time history of the dam's maximum displacement, which can reach 34.4 cm. The maximum displacement is only 22.1 cm when only the mainshock occurs. After the aftershock, the plastic deformations in the structure were further developed, and the maximum displacement reached 34.4 cm. The settlement time history of the upstream dam crest is shown in Figure 18b, and the maximum settlement can reach 70.7 m. Similarly, the maximum settlement is only 37.2 cm when only the mainshock occurs. After the aftershock, the structure with plastic deformation was further damaged, and the maximum settlement reached 70.7 m. Strikingly, compared with the case of a single mainshock, the seismic sequence increases the settlement of the structure by nearly 33.5 cm. Such a large displacement increment will significantly affect the service function of the dam.

Figure 19 illustrates a comparison of the dam's deformation following the mainshock and aftershock, showing that the aftershock obviously exacerbated the damage trend of the dam.

### 4.3. Stochastic Dynamic Results and Reliability

The aftershocks have a significant impact on the dam when the difference $\Delta$ between the dam crest settlement (vertical deformation) after the aftershocks and that after the mainshocks is greater than 6 cm. In 140 groups of dynamic calculations, 33 groups have a difference $\Delta$ greater than 6 cm, as shown in Figure 20. The potential for dams to fail due to seismic sequences is greater than the potential caused by single mainshocks. Therefore, an analysis of the impacts of aftershocks on structures from a probabilistic perspective is needed. Figure 21 displays the average and standard deviation of the structural dynamic response. The means of displacement and settlement (Figure 21a) have similar change characteristics, and the change in the mean of settlement is more obvious than that of displacement after aftershocks. The standard deviations of displacement and settlement

(Figure 21b) have similar characteristics. They changed violently in the first 20 s before and stabilized after 20 s but increased again after 70 s, indicating that aftershocks further increased the dispersion of the response. It is necessary to further extract probability information to study the safety performance of dams under seismic sequences.

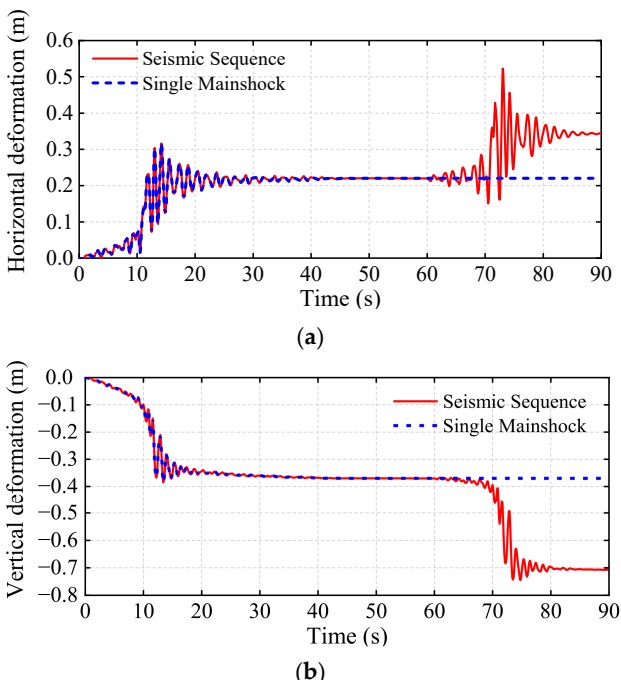

(**a**)

(**b**)

**Figure 18.** Dynamic response time history of single sample. (**a**) Horizontal deformation. (**b**) Vertical deformation.

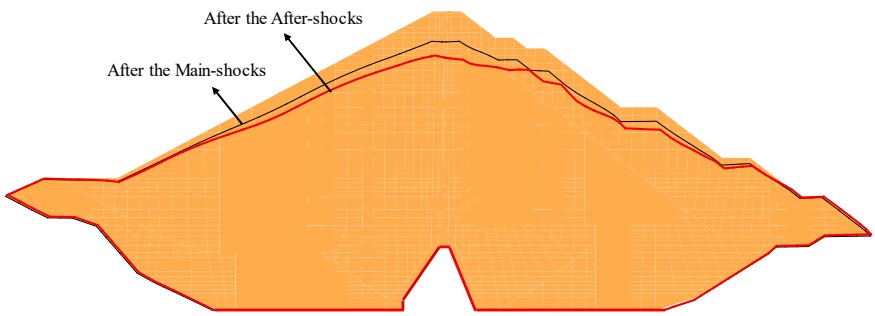

**Figure 19.** Comparison of deformation (amplified 30 times).

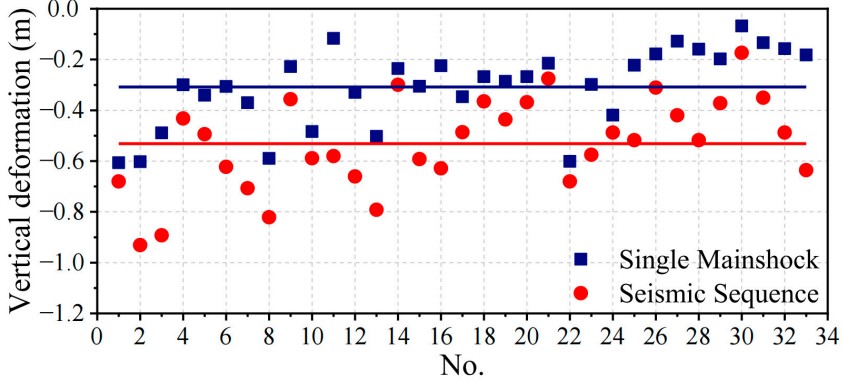

**Figure 20.** Vertical deformation of 34 groups which Δ > 6cm.

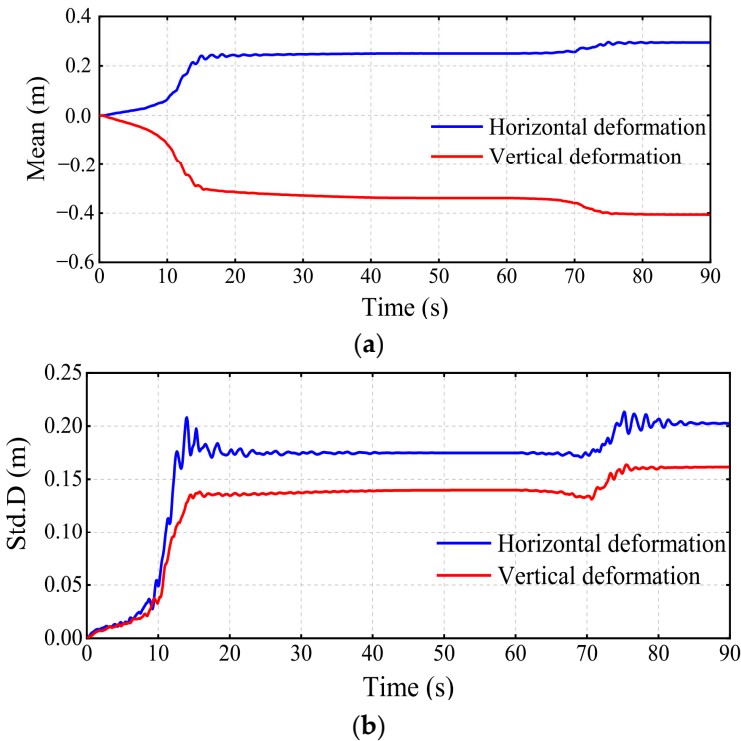

**Figure 21.** Mean and standard deviation of dynamic response. (**a**) Mean of dynamic response. (**b**) Standard deviation of dynamic response.

The evolution process of the PDF of the structural responses can be obtained based on the DPIM, e.g., the PDF surfaces shown in Figure 22. Figure 22a,b show the evolution of displacement PDF over time during the mainshocks of 20–40 s and the aftershocks of 60 s–80 s, respectively. Aftershocks cause greater fluctuations in probability than the later stage of the mainshock, which emphasizes the importance of considering the randomness of aftershocks and its implications. Figure 22c shows the evolution process of the PDF of settlement over 60–80 s. It has similar characteristics to Figure 22a, but fluctuates more violently. This shows that the geotechnical structure is constantly settling under earthquakes, rather than shaking up and down like other elastic structures. Aftershocks still caused more irregular fluctuations, as shown in Figure 22d.

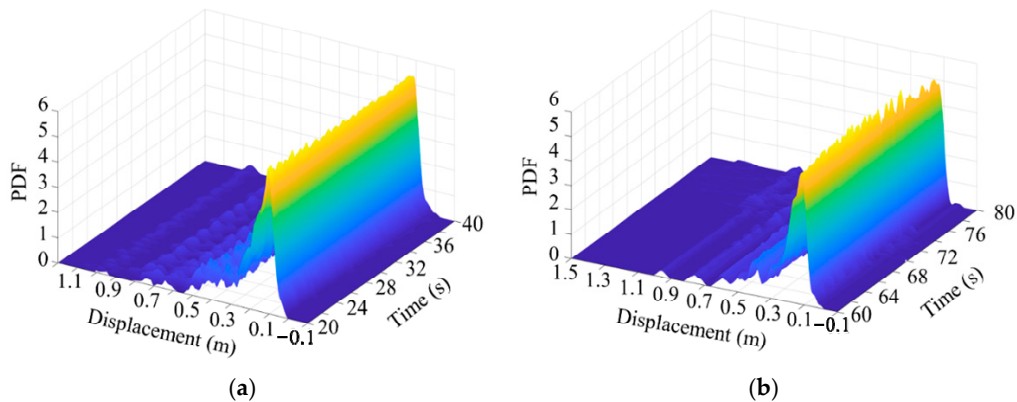

(**a**)  (**b**)

**Figure 22.** *Cont.*

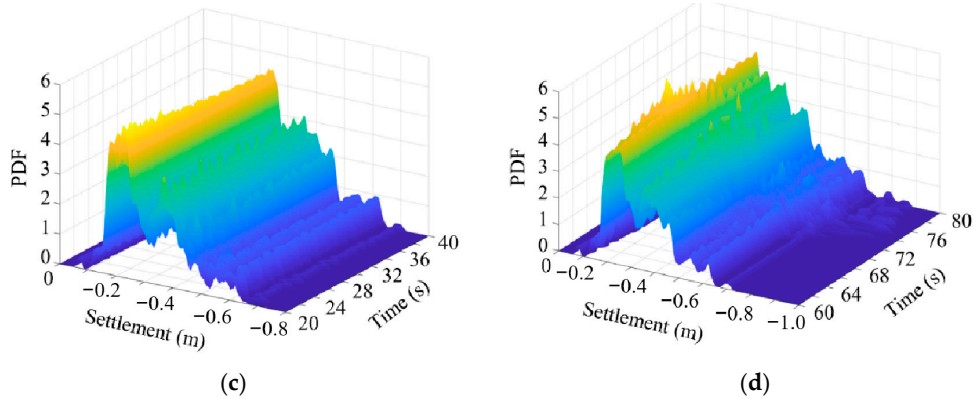

**Figure 22.** PDF surfaces of response at typical times. (**a**) PDF surface of horizontal displacement at 20–40 s. (**b**) PDF surface of horizontal displacement at 60–80 s. (**c**) PDF surface of settlement at 20–40 s. (**d**) PDF surface of settlement at 60–80 s.

The contour of the corresponding PDF surface can also be obtained, as shown in Figure 23. It flows in the time domain, which intuitively reveals the propagation process of random factors $\Theta$. This indicates the liquidity of probability in the state spaces and confirms the conservation of probability from the side aspect.

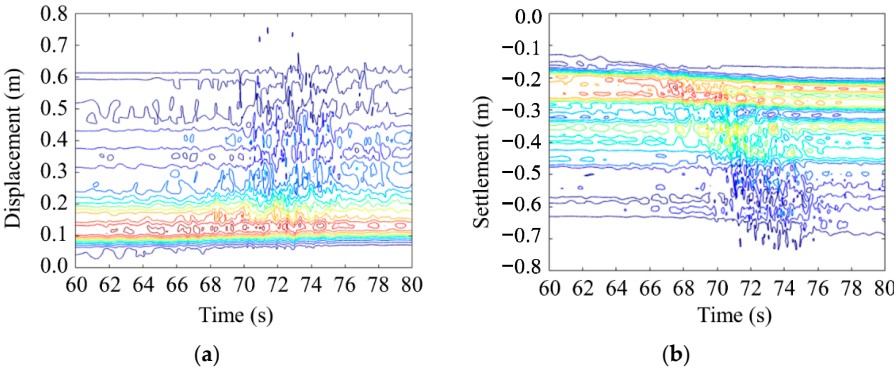

**Figure 23.** PDF contour map at 60–80 s. (**a**) PDF contour of horizontal displacement. (**b**) PDF contour of settlement.

The settlement at the upstream dam crest is a significant indicator when assessing the seismic resistance, and the variability in settlement can be expressed by the PDF at different times, as shown in Figure 24. The distribution of settlements is very concentrated when the seismic peak occurs (PDF at 10 s), while the settlements have a relatively wide distribution after the mainshocks (PDF at 50 s), and the settlements tilt to the more dangerous side after the complete seismic sequence (PDF at 90 s).

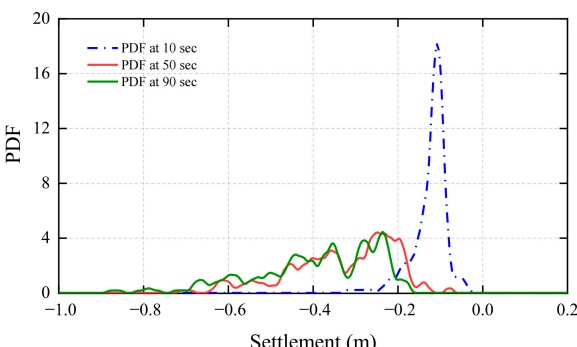

**Figure 24.** PDFs at typical time instants.

The corresponding CDF curves can be easily obtained based on the PDFs. The probability characteristics of each typical time instant can be more clearly reflected by the CDF, as shown in Figure 25.

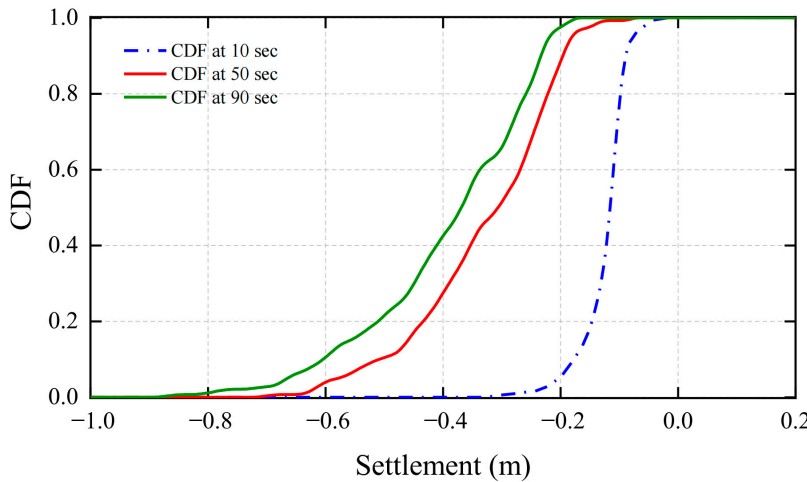

**Figure 25.** CDFs at typical time instants.

The performance CDF can be obtained based on the AC approach proposed in Section 3. Figure 26 shows that no matter how high the threshold is, the reliability in the case of seismic sequences is lower than that in the case of a single mainshock.

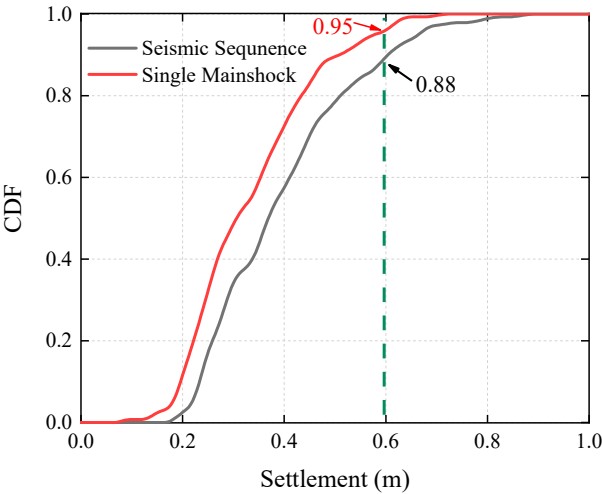

**Figure 26.** CDF of the absorbing condition of permanent settlement.

The first-passage dynamic reliabilities in different thresholds of the dam can be obtained through extreme CDF, as illustrated in Figure 27. The reliability of the structure begins to significantly decrease after the mainshocks, and the reliability will significantly decrease again after the subsequent aftershocks. On the other hand, with the decrease in the threshold, the dynamic reliability of the structure will be greatly reduced, and the time at which the decrease begins will be slightly earlier, as shown in Table 9.

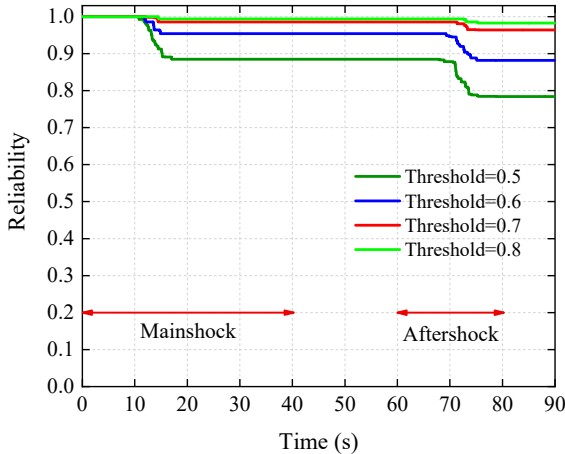

**Figure 27.** Dynamic reliabilities under various thresholds.

**Table 9.** Time when reliability begins to decrease.

| Thresholds | 0.8 m | 0.7 m | 0.6 m | 0.5 m |
|---|---|---|---|---|
| Mainshock | 14.4 s | 12.7 s | 12.0 s | 10.9 s |
| Aftershock | 71.7 s | 69.3 s | 66.9 s | 60.8 s |

To explore the influence of the aftershocks on structural reliability, we focus on the analysis when the threshold is 0.6 m, as shown in Figure 28. The reliability is 0.88 under the single mainshock and further reduces to 0.78 under the seismic sequences. Strikingly, the seismic performance of nonlinear structures shows a decrease in reliability after the peak in the aftershocks, indicating the noticeable impact of aftershocks on structures. In the performance-based seismic design of an earth dam, a reliability of 88% is obviously not enough to consider single mainshock because the aftershocks will significantly reduce the reliability by 10%, making the design or safety evaluation inaccurate.

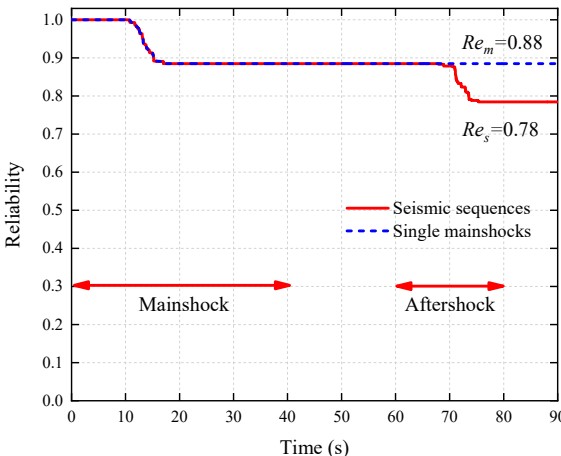

**Figure 28.** Dynamic reliability at threshold = 0.6 m.

In Figure 29, the reliability and difference between a single mainshock and seismic sequences are presented at different thresholds. Apparently, the reliability difference gradually increases with the decrease in threshold; that is, reliability is more likely to significantly decline due to aftershocks in such cases, particularly when the threshold is low. As a consequence, in seismic design that prioritizes performance-based structures, it is essential to consider the impact of aftershocks, especially in situations dealing with non-linear structures where the threshold is low.

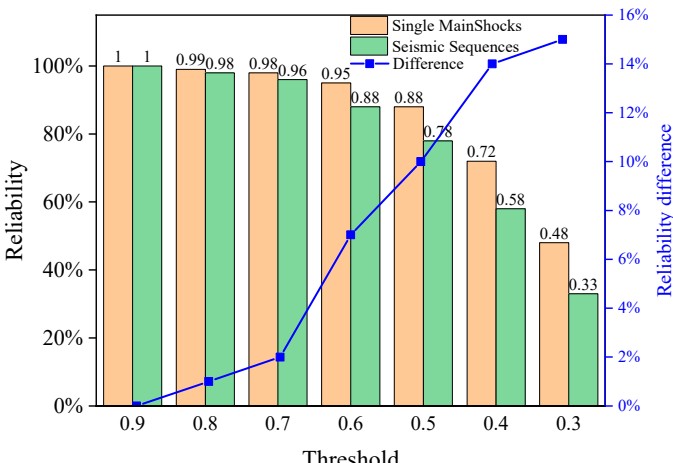

**Figure 29.** Reliability and difference at different thresholds.

## 5. Conclusions

A novel probabilistic approach is proposed for assessing the stochastic dynamic response of earth dams to seismic sequences. In this method, the relevant randomness of mainshocks and aftershocks is introduced, and the decoupling method used to solve the stochastic response is established. This study examines the impact of aftershocks on dam reliability in terms of probability for the first time. The conclusions are as follows:

(1) Simulation method of seismic sequences. This paper proposes a physical model for nonstationary stochastic seismic sequences that employs a source–path–site mechanism, where uncertainties in ground motion are represented as random variables and the source of randomness is identified. The model is evaluated through comparison with actual ground motion data to determine its accuracy.

(2) Generate dependent variables. The multidimensional representative point set with a minimum GFD and different distributions is selected. Utilizing Copula theory, the relationship between mainshocks' and aftershocks' model parameters is established, resulting in more integrated seismic sequences. This approach has a high accuracy with smaller samples in stochastic dynamic analysis.

(3) Stochastic dynamic analysis of the dam. The generalized plasticity theory is introduced into the stochastic dynamic calculation, and the stochastic dynamic response of the dam is solved by DPIM. The displacement-based performance index allows for the calculation of the PDF, CDF, and reliability of a dam at any given time. The study revealed that aftershocks can decrease the reliability of a dam by 1–15%, especially when the design demand is higher. Thus, it is crucial to take aftershocks into account when considering seismic design.

(4) A novel method for a reliability analysis of dams that is both practical and efficient. This identifies the sources of random factors in ground motions and follows the principle of probability conservation. This allows for the comprehensive acquisition of all probability information regarding structural dynamic responses, because it is not limited by different conditions and has strong numerical stability. This method provides a new idea for future engineering design and performance evaluations.

**Author Contributions:** Conceptualization, W.H. and Y.Y.; methodology, M.J.; software, Y.Y.; validation, R.P.; formal analysis, Y.Y.; investigation, R.P.; resources, R.P.; writing—original draft preparation, W.H.; writing—review and editing, Y.Y.; visualization, M.J.; supervision, R.P.; project administration, R.P.; funding acquisition, R.P. All authors have read and agreed to the published version of the manuscript.

**Funding:** This work was supported by National Key R&D Program of China (2021YFB2601102), China National Natural Science Foundation (Grant Nos. 52009017, 51979026, 51890915), Fundamental Research Funds for the Central Universities (DUT21TD106, DUT21RC(3)107), Liaoning Province

Science Foundation (2020-BS-06), and Hebei Key Laboratory of Earthquake Disaster Prevention and Risk Assessment (Grant NO. FZ213201). These financial supports are gratefully acknowledged.

**Data Availability Statement:** The authors do not have permission to share data.

**Conflicts of Interest:** The authors declare that they have no known competing financial interest or personal relationship that could have appeared to influence the work reported in this paper.

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
