# Peer review of "Direct Probability Integral Method for Seismic Performance Assessment of Earth Dam Subjected to Stochastic Mainshock–Aftershock Sequences"

_water, doi:10.3390/w15193485_

Round 1

Reviewer 1 Report

(1) Figures in the manuscript must first be stated in teh text, respectively, and then brief explanations of these figures must be made below the figures

(2) The references in the manuscript must be written strickly in accordance with the journal rules, both in the text and in the references section

Author Response

Dear reviewer #1:
Thank you very much for your decision and constructive comments on our manuscript. We have carefully considered the suggestion of comments and have tried our best to make some adjustments. The improve the manuscript, and revision notes, point-to-point, are given in the resubmitted manuscript file, please feel free to review.

Reviewer 2 Report

Dear Authors, very extended draft and detailed procedures.

Some minor refinements suggested .

Regards

1) Introduce short explanation of passages of procedure after line 183,

2) Improve captions of fig 5 and 6

3) Enlarge simplified description of mathematical procedure at #3.1  and#3.2

4) What is represented in fig.7a and 7b?

5) In case described at #4, give description of vulnerability zones of the structure after repeated seismic actions in terms of continuity and of saturation.

6) Fig. 14 and 15: add a graphical scale.

7) In fig.17, different deformability modulus of materials are not affecting strain effects?

8) Fig 23: which displacements are represented?

Author Response

Dear reviewer #2:
Thank you very much for your decision and constructive comments on our manuscript. We have carefully considered the suggestion of comments and have tried our best to make some adjustments. The improve the manuscript, and revision notes, point-to-point, are given in the resubmitted manuscript file, please feel free to review.
